



# Sensitivity Analysis of Turbine Fatigue and Ultimate Loads to Wind and Wake Characteristics in a Small Wind Farm

Kelsey Shaler[1], Amy Robertson[1], and Jason Jonkman[1]

[1]National Renewable Energy Laboratory, Colorado, USA

**Correspondence:** Kelsey Shaler (kelsey.shaler@nrel.gov)

**Abstract.** Wind turbines are designed using a set of simulations to determine the fatigue and ultimate loads, typically focused solely on unwaked wind turbine operation. These structural loads can be significantly influenced by the wind inflow conditions. When placed in the wake of upstream turbines, turbines experience altered inflow conditions, which can additionally influence the fatigue and ultimate loads. Although significant research and effort has been put into measuring and defining such parameters, limited work has been done to quantify the sensitivity of structural loads to the inevitable uncertainty in these inflow conditions, especially in a wind farm setting with waked conditions. It is therefore important to understand the impact such uncertainties have on the resulting loads of both non-waked and waked turbines. The goal of this work is to assess which wind-inflow- and wake-related parameters have the greatest influence on fatigue and ultimate loads during normal operation for turbines in a three-turbine wind farm. Twenty-eight wind inflow and wake parameters were screened using an elementary effects sensitivity analysis approach to identify the parameters that lead to the largest variation in the fatigue and ultimate loads of each turbine. This study was performed using the National Renewable Energy Laboratory 5 MW baseline wind turbine with synthetically generated inflow based on the International Electrotechnical Commission (IEC) Kaimal turbulence spectrum with IEC exponential coherence model. The focus was on sensitivity to individual parameters, though interactions between parameters were considered, and how sensitivity differs between waked and non-waked turbines. The results of this work show that for both waked and non-waked turbines, ambient turbulence in the primary wind direction and shear were the most sensitive parameters for turbine fatigue and ultimate loads. Secondary parameters of importance for all turbines were identified as yaw misalignment, u-direction integral length, and the exponent and $u$ components of the IEC coherence model. The tertiary parameters of importance differ between waked and non-waked turbines. Tertiary effects account for up to $9.0\%$ of the significant events for waked turbine ultimate loads, and include veer; non-streamwise components of the IEC coherence model; Reynolds stresses; wind direction; air density; and several wake calibration parameters. For fatigue loads, tertiary effects account for up to $5.4\%$ of the significant events, and include vertical turbulence standard deviation; lateral and vertical wind integral lengths; lateral and vertical wind components of the IEC coherence model; Reynolds stresses; wind direction; and all wake calibration parameters. This information shows the increased importance of non-streamwise wind components and wake parameters in fatigue and ultimate load sensitivity of downstream turbines.





# 1  Introduction

When examining the feasibility of a wind farm design for a desired location, simulation models are run to assess the loading that the turbines will encounter given the conditions of that site. These simulation models include a large number of parameters to try to represent the complex conditions the turbine will encounter, and many times much of this wind characterization is not available. It is therefore useful to identify those parameters that have the most significant influence on the load response,

to prioritize measurement campaigns and analysis studies. The focus of this paper is to identify those parameters that have the most influence on the load responses of wind turbines when situated in a farm environment.

This paper builds off our previous work and case studies related to the sensitivity of loads in isolated wind turbines. Our first study focused on assessing the sensitivity of wind inflow parameters on a single turbine (Robertson et al. (2019)). To perform this work, a sensitivity analysis methodology was developed, which employs elementary effects (EE) to provide a sensitivity

estimate, requiring significantly fewer simulations than a full sensitivity analysis. This EE-based sensitivity approach has been employed in all subsequent studies, including the one reviewed in this paper. The single turbine inflow study found that the primary parameters of importance to the fatigue and ultimate loading of the National Renewable Energy Laboratory (NREL)5 MW baseline wind turbine under normal operation were turbulence in the primary wind direction and shear, followed by veer, $u$-direction integral length, and exponent and $u$ components of the IEC coherence model. The second case study

focused on assessing the sensitivity of the aerodynamic parameters of the wind turbine blades, such as lift and drag coefficients as well as unsteady aerodynamic parameters (Shaler et al. (2019)). This study found the primary parameters of importance to be blade twist and lift coefficient distributions (both outboard and inboard), followed by the maximum lift coefficient location, blade chord length, and drag coefficient distributions. The most recent study built upon the blade aerodynamics study to include additional turbine properties such as blade-mass and pitch imbalance, blade and tower center-of-masses, and stiffness

and damping uncertainty, on the wind turbine loads  (Robertson et al. (2019)). This study found the primary parameters of importance to be yaw misalignment and outboard lift coefficient distribution, followed by inboard lift distribution, blade-twist distribution, and blade mass imbalance.

Building off the methods and findings from these previous studies, the work in this paper assesses how waked turbine fatigue and ultimate load sensitivity differs from that of unwaked turbines for varying wind inflow and wake conditions. To accomplish

this, the inflow study that was previously conducted was expanded to include several turbines. Additionally, parameters that effect the wind turbine wake evolution, such as yaw misalignment and model parameters that change wake evolution, were included. An additional wind inflow parameter, air density, was also added. This work aimed to highlight the relative importance of inflow and wake parameters for fatigue and ultimate load sensitivity. This was accomplished by developing metrics to assess the sensitivity of several turbine load measurements, and assessing how this sensitivity changed with varying inflow and wake

conditions. The sensitivity was assessed using the EE method developed in the first case study, considering a wide range of possible wind inflow and wake conditions. From these sensitivity values, a threshold was used to determine when a sensitivity value was classified as a "significant event". From this, the number of "significant events" triggered by varying each parameter were analyzed, along with which aeroelastic quantities of interest (QoI) were most effected. The results from this work can



be used to better inform not only the turbine design process and site-suitability analyses, but also help identify important
measurement quantities when designing wind farm experiments.

## 2  Approach and Methods

To identify the inflow wind and wake parameters that structural loads of waked and unwaked utility-scale wind turbines were
most sensitive to, a sensitivity analysis based on an EE methodology was used. The procedure is summarized in the following
section. However, there are several caveats to this work that must be noted. First, only the NREL 5 MW reference turbine was
considered. Thus, this study does not examine the dependency of the sensitivity findings on the size and design of the turbine.
Secondly, only normal turbine operation was considered. Gusts, start-ups, shutdown, and parked or idling events were not
included, which can often lead to the high loading experienced by a turbine. And finally, input parameter variation was done
independently, with no joint-probability functions or conditioning based on any parameter other than wind speed. Developing
joint-probability distributions across the large number of parameters considered was not feasible. Despite these caveats, this
work still provides insight into the sensitivity of fatigue and ultimate loads based on the variation of a wide range of wind
inflow and wake conditions.

### 2.1  Wind Turbine Model and Tools

The sensitivity study was performed considering a small wind farm with three laterally aligned NREL 5 MW reference wind
turbines (Jonkman et al. (2009)) separated by 7 rotor diameters in the zero-degree wind direction, as shown in Figure 1.
Parameter sensitivity was assessed using simulations from FAST.Farm, a multi-physics engineering tool that accounts for
wake interaction effects on turbine performance and structural loading in wind farm applications. FAST.Farm is an extension
of the NREL software OpenFAST, which solves the aero-hydro-servo-elasto dynamics of individual turbines (Jonkman and
Shaler (2020)).

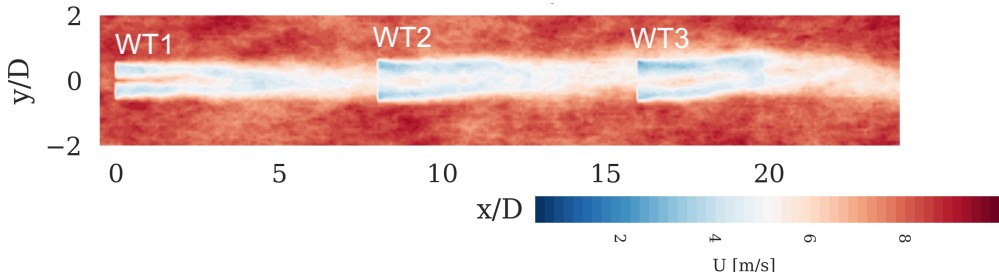

**Figure 1.** Instantaneous 2D flow visualization of a three-turbine FAST.Farm simulation in turbulent inflow, sampled at hub height and colored
by velocity magnitude.

Each wind turbine was modeled in OpenFAST, using the NREL 5 MW reference turbine as a representative turbine. This is
an upwind three-bladed horizontal-axis turbine with a 90 m hub height, and 126 m rotor diameter. *AeroDyn*, the aerodynamic



module of OpenFAST, was applied to calculate the aerodynamic loads on the rotor using blade-element momentum (BEM) theory with advanced corrections, including unsteady aerodynamics. *ElastoDyn*, a combined multi-body and modal structural approach that includes geometric nonlinearities, was used to represent the flexibility of the blades, drivetrain, and tower. Tower influence on the flow and nacelle blockage, as well as drag on the tower, were not considered. The NREL 5 MW turbine baseline
controller was modeled as a variable-speed collective pitch controller using a Bladed-style dynamic library in *ServoDyn*. OpenFAST simulation results were used to compute the EE values for each QoI, as discussed in Section 3.

Past work has shown that the sensitivity of loads to input parameter variation is influenced by the wind speed and associated wind turbine controller response (Robertson et al. (2019)). Therefore, this study considered three different wind speeds at mean hub height wind speeds of 8, 12, and 18 m/s, representing below-, near-, and above-rated wind speeds, respectively. Wind
inflow was synthetically generated using TurbSim (Jonkman (2014)), which creates time-varying two-dimensional turbulent flow fields that are convected through the domain using Taylor's frozen turbulence hypothesis. Turbulence was simulated using the Kaimal turbulence spectrum with an exponential coherence model. TurbSim generation involves two stages of simulations, one each for the low-resolution and high-resolution domains of FAST.Farm and using the suggested FAST.Farm discretization recommendation (Jonkman and Shaler (2020)). The low-resolution TurbSim domain throughout the wind farm had a spatial
resolution of 10, 20, and 25 m for the below-, near-, and above-rated wind speeds, respectively, and a temporal resolution of 0.1 seconds to match the suggested high-resolution FAST.Farm discretization. A high-resolution TurbSim domain around each wind turbine was then generated for each turbine, derived from the hub height time series extracted from the low-resolution TurbSim domain with a spatial resolution of 5 m and temporal resolution of 0.1 seconds. Many turbulence seeds were used for each input parameter variation to ensure any variation in results was independent of the selected turbulent seed. The number
of seeds was determined via a seed convergence study that considered each QoI. The generated inflows were used as input to FAST.Farm using a simulation time of 600 seconds after an initial 600 seconds transient period was removed.

## 2.2  Case Study

In previous case studies (Robertson et al. (2019)), ambient wind-inflow parameters were identified that significantly influence the loading of a single wind turbine. This study extends that work to identify the inflow and wake parameters most influencing
downstream wind turbines in a small wind farm. The ambient wind inflow input parameters were selected to be the same ones used in our previous work (Robertson et al. (2019)). Additional wake parameters were added that relate to turbine wake development/meandering. Though more parameters could exist, for this study only those parameters believed to have the largest effect for normal operation for a conventional utility-scale wind turbine were included, as categorized in Figure 2.

Many QoIs were identified, as detailed in Table 1, include the blade, tower, and drivetrain moments; blade-tip displacement;
rotor power; and inflow turbulence intensity (TI) of each turbine. Inflow TI was computed using Equation 1 (where $t$ is the turbine number) and not as a fatigue load.

$$TI_t = \frac{\sigma(u)}{\overline{u}} \qquad (1)$$





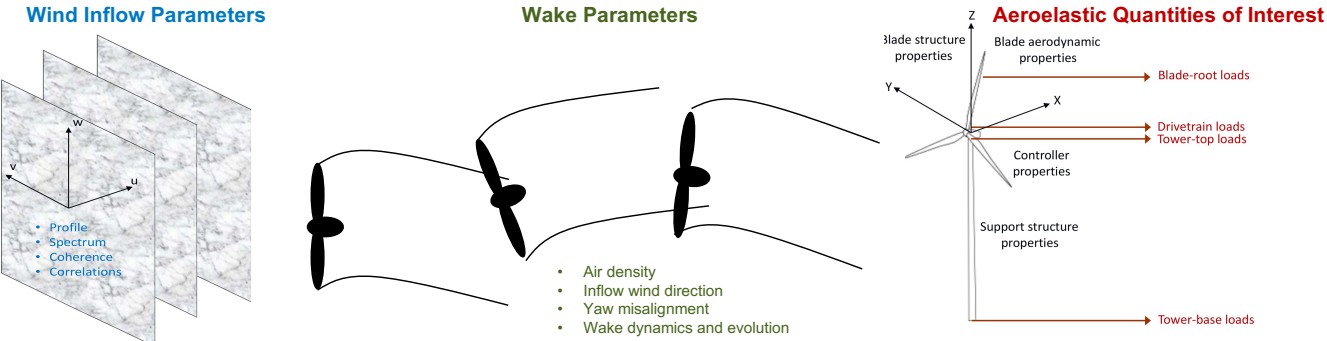

**Figure 2.** Potential sources of uncertainty in a wind turbine and wind farm load analysis. Includes wind-inflow conditions; wake parameters; and the associated load QoIs.

**Table 1.** Quantities of interest and relevant vector components.

| Quantity of Interest | Component | | |
|---|---|---|---|
| Blade-root moments | Out-of-plane (OoP) bending | In-plane (IP) bending | Pitching moment |
| Low-speed shaft moment at main bearing | $0°$ bending | $90°$ bending | Shaft torque |
| Tower-top moments | Fore/aft (FA) bending | Side/side (SS) bending | Yaw moment |
| Tower-base moment | FA bending | SS bending | |
| Blade-tip displacements | OoP (ultimate only) | | |
| Electrical power | WT1, 2, 3 separately | WT1, 2, 3 summed (ultimate only) | |
| Inflow TI | WT1, 2, 3 (fatigue only) | | |

The fatigue loads were calculated using aggregate damage-equivalent loads (DEL) of the QoI response across all turbulence seeds for a given set of short-term parameter values. For the bending moments, the ultimate loads were calculated as the largest vector sum of the first two listed components. The ultimate loads were calculated using the average of the global absolute maximums across all turbulence seeds for a given set of parameter values. See Robertson et al. (2019) for more details on the fatigue and ultimate loads calculations. All quantities associated with electrical power and inflow TI were excluded from the significant event count, but were examined for other purposes. The QoI sensitivity of each input parameter was examined using the procedure summarized in Section 3.

## 3  Elementary Effects Procedure

An EE method (Gan et al. (2014); Martin et al. (2016); Saranyasoontorn and Manuel (2006)) was used to assess which parameters have the largest influence on turbine loads. This is a simple methodology for screening parameters, based on a one-at-a-time approach where each parameter is varied independently while all other parameters remain fixed. In this way, the EE method





is a local sensitivity approach because the influence of a single parameter is calculated without considering interaction with

125 other parameters. The change in response QoIs based on the change in the input parameter was used to compute a derivative, which together with the possible range of the input parameter variation was used to assess the sensitivity of the parameter. This variation and derivative computation was performed several times for each parameter at different points in the hyperspace of all input parameters, as shown in Figure 3. In this way, the EE approach used in this work is considered a global sensitivity method because it concerns the interactions between different parameters (Robertson et al. (2019)). This method and evaluation

process are further discussed by Robertson et al. (2019).

When considering the EE method, each wind turbine QoI, $Y$, was represented as a function of different characteristics of the inflow and wake input parameters, $\mathbf{U}$, as follows:

$$Y = f(u_1, \ldots, u_i, \ldots, u_I), \tag{2}$$

where $I$ is the total number of input parameters. For a given sampling of $\mathbf{U}$, the EE value of the $i$th input parameter was found by varying only that parameter by a normalized amount, $\Delta$:

$$EE_i = \frac{f(\mathbf{U} + \mathbf{x}_k) - f(\mathbf{U})}{\Delta} \tag{3}$$

where

$$
\quad \mathbf{x}_k =
\begin{cases} 
0 & \text{for } k \neq i, \\ 
\Delta & \text{for } k = i. 
\end{cases} \tag{4}
$$

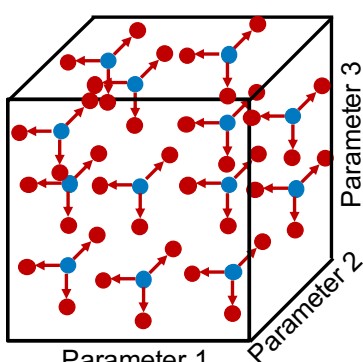

**Figure 3.** Radial EE approach representation for three input parameters. Blue circles indicate starting points in the parameter hyperspace. Red points indicate variation in one parameter at a time.

Because of the normalization of $\mathbf{U}$, the EE value ($EE_i$) can be thought of as the local partial derivative of the output ($Y$) with respect to an input ($u_i$), scaled by the range of that input. Thus, the EE value has the same unit as the output QoI.

In a radial sensitivity approach, the EE value is calculated for all input parameters at a given point, $R$, in the parameter hyperspace by varying each parameter individually from that point. A representative schematic of this approach is depicted

in Figure 3. Each variation is performed for $\pm 10\%$ of the range over which the parameter may vary ($\Delta = \pm 0.1$). This $\pm 10\%$ range ($\Delta = \pm 0.1$ normalized or $\Delta_{ib} = \pm 0.1 u_{ib,range}$ dimensional) is used to ensure the finite difference calculation occurs over an appropriate range to meet the linearity assumption required by this method. Note that this is different than the original EE methodology, which creates a trajectory by varying each new parameter from the $\Delta$ point of the previous parameter. This process is repeated for $R$ starting points in the input parameter hyperspace (blue points in Figure 3), creating a set of $R$

different calculations of EE value for each parameter. The $R$ starting points area determined using Sobol numbers (blue circles in Figure 3), which ensures a wide sampling of the input hyperspace.

Because EE value is analogous to a sensitivity level, a higher value for a given input parameter indicates more sensitivity. Here, the most sensitive parameters were identified by defining a threshold value, above which an individual EE value would be considered significant. The threshold was set individually for each QoI and turbine and defined as $\overline{EE}^r + 2\sigma$. Here, $\overline{EE}^r$



**Table 2.** Wind inflow and wake parameters.

| Mean Wind Profile | Velocity Spectrum | Spatial Coherence | Component Correlation | Wake Parameters |
|---|---|---|---|---|
| Shear ($\alpha$) | Standard deviation ($\sigma_u, \sigma_v, \sigma_w$) | Input coherence decement ($a_u, a_v, a_w$) | Reynolds Stresses ($PC_{uv}, PC_{uw}, PC_{vw}$) | Wind direction ($WD$) |
| Veer ($\beta$) | Integral scale parameter ($L_u, L_v, L_w$) | Offset parameter ($b_u, b_v, b_w$) | | Air Density ($\rho$) |
| | | Exponent ($\gamma$) | | Yaw misalignment ($\Theta_{T1}, \Theta_{T2}, \Theta_{T3}$) |
| | | | | Wake calibration parameters ($C_{NearWake}, C_{Meander}, k_{\nu Amb}, k_{\nu Shr}, f_c$) |

was the mean of all EE values across all starting points $R$, inputs $I$, and wind speed bins $B$ for each QoI and $\sigma$ was the standard deviation of these EE values.

### 3.1 Parameters

A total of 28 input parameters represented the wind-inflow and wake conditions, considering the mean wind profile, velocity spectrum, spatial coherence, component correlation, and wake parameters, as summarized in Table 2. The wind inflow pa-

160 rameters were detailed in previous work (Robertson et al. (2019)). Wind direction was used to essentially introduce lateral offset distances for downstream turbines, with zero degrees direction indicating flow directly down the row of turbines. Wind direction was simulated by changing the locations of the wind turbines in the FAST.Farm simulations. This way, the same inflow turbulence files could be used for various wind direction values. Changing the wind direction does not result in a mean yaw misalignment of the wind turbines. Air density was specified in *AeroDyn* and represents the change due to temperature or

165 humidity variations. Yaw misalignment was specified by rotating the nacelle-yaw angle of each wind turbine individually in *ElastoDyn*. Wake calibration parameters were FAST.Farm user-specified parameters that modify wake dynamics evolution and meandering. $C_{NearWake}$ adjusts the wake deficit and expansion correction for the otherwise neglected pressure gradient zone directly behind the rotor in the near wake. $C_{Meander}$ influences the spatial averaging used to calculate how the wake meanders and specifically defines the cut-off wave number for the spatial filter. $k_{\nu Amb}$ and $k_{\nu Shr}$ modulate the relative contribution of the

170 ambient turbulence and wake shear layer to the eddy viscosity. $f_c$ defines the cutoff frequency for the low-pass time filter used in the wake evolution model to ensure high-frequency fluctuations do not pass into the quasi-steady wake-deficit increment model.

To understand the sensitivity of a given parameter, a range over which that parameter may vary must be defined, as summarized in Tables 3 and 4. A literature search was done to identify the range for each of the parameters across varying onshore

installation sites. When possible, parameter ranges were set based on wind speed bins. If no information on wind speed





dependence was found, the same values were used for all bins. Many of these ranges were chosen based on our previous study (Robertson et al. (2019)). Air density ranges were based on the work of (Ulazia et al. (2019)) and represents the changes due to temperature or humidity variations. The wind direction was chosen based on the work of Gaumond et al. (2014), which looked at wind direction uncertainty in experimental measurements. Yaw misalignment ranges were based on the work

of Quick et al. (2017). For the wake parameters, ranges were chosen based on a calibration study used to determine the default FAST.Farm parameters (Doubrawa et al. (2018)).

## 4   Results

The EE value was calculated for each of the 28 input parameters ($I$) at 30 different starting points ($R$) in the input parameter hyperspace. The number of starting points was determined through a convergence study on the average EE value. At each

185 of the considered points, 50 TurbSim simulations for 50 turbulence seeds ($S$)were run. The number of turbulence seeds was determined based on a convergence study of the fatigue and ultimate load metrics at the mid-point range value and $\pm 10\%$ of the range for all QoIs. Based on these parameters, the total number of TurbSim simulations performed was $R \times (J + 1) \times S \times B = 30 \times 19 \times 50 \times 3 = 85,500$, where $J$ was the number of input parameters that required a new TurbSim simulation and $B$ was the number of wind speed bins considered. The total number of FAST.Farm simulations performed was $R \times (I + 1) \times S \times B = $

$30 \times 29 \times 50 \times 3 = 130,500$.

To demonstrate how EE values can vary for a given input parameter and QoI, ordered EE value results of blade-root pitching fatigue and ultimate loads are shown in Figure 4. Here, input parameters were plotted independently of each other to compare the behavior between parameters. Each line represents a different input parameter, with each point representing a different location in the hyperspace. Additionally, each sub figure represents a different wind speed bin and each line color represents

a different wind turbine. For each line, the EE values for each point were ordered from smallest to largest and the point was assigned a value from 1 to 30, one for each starting point in the hyperspace, corresponding to the y-axis value in the figure. The vertical lines on each plot correspond to the threshold value, used to identify significant events. Markers above this threshold line were included in the significant events tally, discussed next. From these plots, it was seen that the shear exponent heavily dominates the results, especially for the below-rated wind conditions. This was seen in previous work (Robertson et al. (2019))

and was largely due to the sizeable range considered for this value. These plots also demonstrate the differences in EE values across wind turbines. For instance, the maximum EE value for ultimate loads at below-rated was due to the shear parameter for all turbines. However, this EE value was 32% higher for WT3 compared to the value for WT1, thus demonstrating the potential differences in parameter importance for waked conditions.

To identify the most sensitive parameters, a tally was made of the number of times an EE value exceeded the threshold for

each QoI. The resulting tallies are shown in Figure 5, with the ultimate load tally on the left (a, c) and the fatigue load tally on the right (b, d). The top figures (a, b) show the cumulative values for each turbine. These results indicate substantial sensitivity to the u-direction turbulence standard deviation ($\sigma_u$) and vertical wind shear ($\alpha$) for all wind turbines. These results were expected based on our past studies (Robertson et al. (2019)) and the parameters of importance in the IEC design standards.



**Table 3.** Wind inflow parameter ranges separated by wind speed bin. The nominal value prescribed by IEC for category B turbulence is specified in the "IEC" row.

| Variable | $\alpha$ | $\beta$ | $L_u$ | $L_v$ | $L_w$ | $\sigma_u$ | $\sigma_v$ | $\sigma_w$ | $a_u$ | $a_v$ | $a_w$ | $b_u$ | $b_v$ | $b_w$ | $\gamma$ | $PC_{uv}$ | $PC_{uw}$ | $PC_{vw}$ |
|---|---|---|---|---|---|---|---|---|---|---|---|---|---|---|---|---|---|---|
| Units | (-) | (°) | (m) | (m) | (m) | (m/s) | (m/s) | (m/s) | (-) | (-) | (-) | (m$^{-1}$) | (m$^{-1}$) | (m$^{-1}$) | (-) | (m$^2$s$^{-2}$) | (m$^2$s$^{-2}$) | (m$^2$s$^{-2}$) |
| **Below-Rated Wind Speeds** | | | | | | | | | | | | | | | | | | |
| IEC | 0.2 | 0 | 340 | 110 | 28 | 1.6 | 1.3 | 0.8 | 12 | - | - | 3.5E-4 | - | - | 0 | - | - | - |
| Min. | -0.75 | -25 | 5 | 2 | 2 | 0.05 | 0.02 | 0.03 | 1.5 | 1.7 | 2 | 0 | 0 | 0 | 0 | -3.5 | -4.5 | -2.7 |
| Max. | 3.3 | 50 | 1000 | 1000 | 650 | 7.2 | 7.4 | 4.5 | 26 | 18 | 17 | 0.08 | 4.5E-3 | 0.011 | 1 | 0.5 | 6.0 | 1.0 |
| **Near-Rated Wind Speeds** | | | | | | | | | | | | | | | | | | |
| IEC | 0.2 | 0 | 340 | 110 | 28 | 2.0 | 1.6 | 1.0 | 12 | - | - | 3.5E-4 | - | - | 0 | - | - | - |
| Min. | -0.4 | -10 | 8 | 2 | 2 | 0.2 | 0.05 | 0.05 | 1.5 | 1.7 | 2 | 0 | 0 | 0 | 0 | -3.5 | -4.5 | -2.7 |
| Max. | 0.9 | 50 | 1400 | 1300 | 450 | 7.3 | 8.1 | 4.3 | 26 | 18 | 17 | 0.08 | 3.0E-3 | 6.0E-3 | 1 | 0.5 | 6.0 | 1.0 |
| **Above-Rated Wind Speeds** | | | | | | | | | | | | | | | | | | |
| IEC | 0.2 | 0 | 340 | 110 | 28 | 2.7 | 2.1 | 1.4 | 12 | - | - | 3.5E-4 | - | - | 0 | - | - | - |
| Min. | -0.4 | -10 | 25 | 2 | 2 | 0.2 | 0.18 | 0.15 | 1.5 | 1.7 | 2 | 0 | 0 | 0 | 0 | -3.5 | -4.5 | -2.7 |
| Max. | 0.7 | 25 | 1600 | 1500 | 650 | 7.4 | 7.3 | 4.2 | 26 | 18 | 18 | 0.05 | 2.5E-3 | 6.5E-3 | 1 | 0.5 | 6.0 | 1.0 |

**Table 4.** Wake parameter ranges separated by wind speed bin.

| Variable | Wind Direction | $\rho$ | $\Theta_{T1}$ | $\Theta_{T2}$ | $\Theta_{T3}$ | $C_{NearWake}$ | $k_{vAmb}$ | $k_{vShr}$ | $C_{Meander}$ | $f_c$ |
|---|---|---|---|---|---|---|---|---|---|---|
| Units | (°) | (kg/m$^3$) | (°) | (°) | (°) | (-) | (-) | (-) | (-) | (-) |
| **Below-Rated Wind Speeds** | | | | | | | | | | |
| Min. | -10.0 | 1.1393 | -20.0 | -20.0 | -20.0 | 1.2 | 0.01 | 0.01 | 1.3 | 0.0001 |
| Max. | 10.0 | 1.3108 | 20.0 | 20.0 | 20.0 | 2.20 | 0.09 | 0.02 | 2.20 | 0.001 |
| **Near-Rated Wind Speeds** | | | | | | | | | | |
| Min. | -10.0 | 1.1393 | -20.0 | -20.0 | -20.0 | 1.2 | 0.01 | 0.01 | 1.3 | 0.0001 |
| Max. | 10.0 | 1.3108 | 20.0 | 20.0 | 20.0 | 2.20 | 0.09 | 0.02 | 2.20 | 0.0015 |
| **Above-Rated Wind Speeds** | | | | | | | | | | |
| Min. | -10.0 | 1.1393 | -20.0 | -20.0 | -20.0 | 1.2 | 0.01 | 0.01 | 1.3 | 0.0001 |
| Max. | 10.0 | 1.3108 | 20.0 | 20.0 | 20.0 | 2.20 | 0.09 | 0.02 | 2.20 | 0.0023 |



(a) Ultimate Loads



(b) Fatigue Loads

**Figure 4.** Exceedance probability plot of (a) ultimate and (b) fatigue load EE values for blade-root pitching moments. Each subplot shows a different wind speed bin. Each line represents a different input parameter and wind turbine (blue for WT1, green for WT2, and yellow for WT3). Each symbol represents a different point in the hyperspace. The vertical lines on each plot correspond to the threshold value used to identify significant events. Y-axis has been limited to focus on results contributing to significant event count.





(a) Ultimate Loads Tally

(b) Fatigue Loads Tally

(c) Relative Ultimate Loads

(d) Relative Fatigue Loads

**Figure 5.** Significant parameter count from ultimate (left) and fatigue (right) loads. Each color represents a different wind turbine. The top row (a and b) shows the significant event counts and the bottom row (c and d) shows the percent difference in significant event counts for WT2 and WT3 relative to WT1. For the $\Theta$ input parameters, the values in (c) and (d) can extend to nearly $\pm 100\%$. However, the axis of these figures have been reduced to better focus on the impact of more input parameters.

Considering the lower tally values in this plot highlights the secondary level of importance of yaw misalignment ($\Theta_{T1,T2,T3}$), streamwise u-direction integral length ($L_u$), u-direction components of the IEC coherence model ($a_u$ and $b_u$), and the IEC coherence model exponent ($\gamma$). As expected, wake calibration parameters have no effect on the unwaked turbine, but do appear with significant events for the waked turbines. Additional insights shown here that were not seen in the previous study was the changing effect of yaw misalignment for downstream turbines. Results for each turbine show high sensitivity to that turbine's yaw misalignment. However, there was little to no dependence on the yaw misalignment of other turbines. It was expected





that the yaw misalignment of a downstream turbine would not effect an upstream turbine result, but less expected that the
reverse was not also true; i.e., that the yaw misalignment of an upstream turbine has little to no effect on the sensitivity of
the turbine directly downstream of it considering recent work on wake steering in the wind energy community. There was a
slight effect of $\Theta_{T2}$ on WT3, but this effect was minimal, especially relative to the effect of $\Theta_{T3}$ on WT3. The primary and
secondary importance parameters were the same for fatigue and ultimate loads, as well as for each turbine, with WT1 results
being consistent with the results in Robertson et al. (2019).

$$\text{Diff}_{t,i} = \left( \frac{\text{SigCount}_{i,WT_t}}{\sum_i \text{SigCount}_{i,WT_t}} - \frac{\text{SigCount}_{i,WT_1}}{\sum_i \text{SigCount}_{i,WT_1}} \right) \times 100 \tag{5}$$

However, the relative importance of these parameters between fatigue and ultimate loads and between wind turbine does
change, as shown in Figures 5(c) and 5(d). Here, the differences between the waked and non-waked turbine response were
explored by showing the difference in the percentage a certain parameter makes up of the total number of significant event
counts for that turbine, relative to WT1. These values were computed using Equation 5, where $t = 2$ or 3, and $i$ was the input
parameter being varied. The percent difference results show when input parameters lead to a higher or lower percentage of
significant events counts in waked turbines, relative to the non-waked turbine. For ultimate loads, WT2 and WT3 show reduced
sensitivity for many of the input parameters, but also increased sensitivity for parameters that show little to no significance for
the non-waked turbine, such as lateral wind components and wake parameters. Similar results were seen for fatigue results.
From here, tertiary effects can be identified for waked turbines. Tertiary effects for ultimate loads show the importance of veer
($\beta$), non-streamwise components of the IEC coherence model ($a_w$ and $b_v$), Reynolds stresses ($PC_{uv}$, $PC_{uw}$, and $PC_{vw}$), wind
direction ($WD$), air density ($\rho$), and several wake calibration parameters ($C_{Meander}$, $k_{\nu,Amb}$, $k_{\nu,Shr}$, and $f_c$). For WT1 and
WT2, these tertiary parameters accounted for 3.2% and 3.6% of the total significant events count, respectively, and nearly
triple that for WT3, with 9.0% of the significant events resulting from tertiary parameters. This suggests that the importance of
these other parameters would likely grow if additional wind turbines where added to the wind farm. Tertiary effects for fatigue
loads show the importance of vertical turbulence standard deviation $\sigma_w$, lateral and vertical wind integral lengths ($L_v$ and $L_w$),
lateral and vertical wind components of the IEC coherence model ($a_w$, $b_v$, and $b_w$), Reynolds stresses ($PC_{uw}$ and $PC_{vw}$),
wind direction ($WD$), and all wake calibration parameters ($C_{NearWake}$, $C_{Meander}$, $k_{\nu,Amb}$, $k_{\nu,Shr}$, and $f_c$). For WT1, these
tertiary parameters accounted for 4.1% of the total significant events count. For WT2 and WT3, this percentage increases to
5.4% and 5.3% of the significant events, respectively. These results indicate the increased influence of non-streamwise inflow
components, including wake meandering, in fatigue and ultimate loads sensitivity of waked turbines.

    This point was further made by comparing the percentage of contribution to the total number of significant event for fatigue
and ultimate loads for each turbine, shown in Figure 5. These results show that most of the tertiary parameters contribute at least
twice as much to the significant events count for waked turbines, compared to unwaked turbines. This indicates that, though
still tertiary parameters, fatigue and ultimate loads of waked turbines were generally twice as sensitive to non-streamwise
inflow components.





**Table 5.** Percentage of contribution to total number of significant events for fatigue and ultimate loads. Cells are colored by the percentage value, with darker blue representing a higher percentage.

(a) Ultimate Loads

| | $\alpha$ | $\beta$ | $\sigma_u$ | $\sigma_v$ | $\sigma_w$ | $L_u$ | $L_v$ | $L_w$ | $a_u$ | $a_v$ | $a_w$ | $b_u$ | $b_v$ | $b_w$ | $\gamma$ | $PV_{uw}$ | $PV_{uv}$ | $PV_{vw}$ | WD | $\rho$ | $\Theta_{T1}$ | $\Theta_{T2}$ | $\Theta_{T3}$ | $C_{NW}$ | $k_{vAmb}$ | $k_{vShr}$ | $C_M$ | $f_c$ |
|---|---|---|---|---|---|---|---|---|---|---|---|---|---|---|---|---|---|---|---|---|---|---|---|---|---|---|---|---|
| WT1 | 28.97 | 0.51 | 31.77 | 2.03 | 0.51 | 2.03 | 2.16 | 1.27 | 10.04 | 0.89 | 0.76 | 8.26 | 0.38 | 0.25 | 3.56 | 0.76 | 0.25 | 0.25 | 0.00 | 0.25 | 5.08 | 0.00 | 0.00 | 0.00 | 0.00 | 0.00 | 0.00 | 0.00 |
| WT2 | 31.57 | 0.13 | 33.55 | 1.98 | 0.13 | 2.25 | 1.32 | 0.66 | 9.25 | 0.40 | 0.66 | 8.45 | 0.26 | 0.13 | 3.57 | 2.11 | 0.13 | 0.13 | 0.00 | 0.13 | 0.00 | 3.17 | 0.00 | 0.00 | 0.00 | 0.00 | 0.00 | 0.00 |
| WT3 | 27.94 | 1.01 | 31.48 | 0.88 | 0.51 | 2.78 | 0.76 | 0.76 | 9.61 | 0.88 | 1.39 | 7.21 | 0.76 | 0.25 | 4.68 | 1.14 | 1.39 | 0.76 | 0.63 | 0.38 | 0.38 | 0.51 | 2.40 | 0.00 | 0.38 | 0.25 | 0.51 | 0.38 |

(b) Fatigue Loads

| | $\alpha$ | $\beta$ | $\sigma_u$ | $\sigma_v$ | $\sigma_w$ | $L_u$ | $L_v$ | $L_w$ | $a_u$ | $a_v$ | $a_w$ | $b_u$ | $b_v$ | $b_w$ | $\gamma$ | $PV_{uw}$ | $PV_{uv}$ | $PV_{vw}$ | WD | $\rho$ | $\Theta_{T1}$ | $\Theta_{T2}$ | $\Theta_{T3}$ | $C_{NW}$ | $k_{vAmb}$ | $k_{vShr}$ | $C_M$ | $f_c$ |
|---|---|---|---|---|---|---|---|---|---|---|---|---|---|---|---|---|---|---|---|---|---|---|---|---|---|---|---|---|
| WT1 | 19.47 | 0.96 | 43.10 | 2.08 | 0.40 | 2.63 | 1.52 | 0.64 | 6.46 | 0.88 | 0.08 | 6.30 | 0.56 | 0.40 | 6.46 | 0.24 | 0.56 | 0.24 | 0.00 | 0.24 | 6.78 | 0.00 | 0.00 | 0.00 | 0.00 | 0.00 | 0.00 | 0.00 |
| WT2 | 19.54 | 0.74 | 42.03 | 1.72 | 0.16 | 1.96 | 1.55 | 0.82 | 7.44 | 0.90 | 0.49 | 6.30 | 0.74 | 0.65 | 7.36 | 0.57 | 0.41 | 0.16 | 0.08 | 0.16 | 0.08 | 5.97 | 0.00 | 0.08 | 0.00 | 0.00 | 0.00 | 0.08 |
| WT3 | 19.52 | 0.74 | 43.01 | 1.74 | 0.50 | 1.99 | 1.08 | 0.91 | 7.61 | 0.58 | 0.41 | 7.53 | 0.25 | 0.66 | 6.62 | 0.33 | 0.33 | 0.41 | 0.33 | 0.08 | 0.08 | 0.08 | 4.80 | 0.08 | 0.08 | 0.08 | 0.08 | 0.08 |

Histogram plots of blade-root pitching moment EE values are shown in Figures 6 and 7 for ultimate and fatigue loads, respectively. These figures show EE value histograms showing the contribution from all input parameters, with wind speed bins and turbines shown in separate subplots. Here, tertiary parameters are highlighted in bright colors to better recognize when they contribute to the significant event count. For ultimate loads, the distribution of outliers were consistent across the turbines, with most outliers occurring at below-rated wind speeds. Tertiary effects do, however, occur the most for WT3, in particular at above-rated wind speeds. Similar results were seen for the fatigue load results in Figure 7, with overall distributions remaining consistent across the turbines, but tertiary effects occurring the most for WT3 and near-rated wind speeds.



**Figure 6.** Histograms of ultimate load EE values for the blade-root pitching moment. Each subplot shows one wind speed bin and wind turbine and includes all input parameters. Each column of subplots corresponds to a wind speed bin and each row of subplots corresponds to a wind turbine. The vertical lines on each plot correspond to the threshold value used to identify significant events. Y-axis has been limited to focus on results contributing to significant event count.



**Figure 7.** Histograms of fatigue load EE values for the blade-root pitching moment. Each graph shows one wind speed bin and wind turbine and includes all input parameters. Each column of subplots corresponds to a wind speed bin and each row of subplots corresponds to a wind turbine. The vertical lines on each plot correspond to the threshold value used to identify significant events. Y-axis has been limited to focus on results contributing to significant event count.





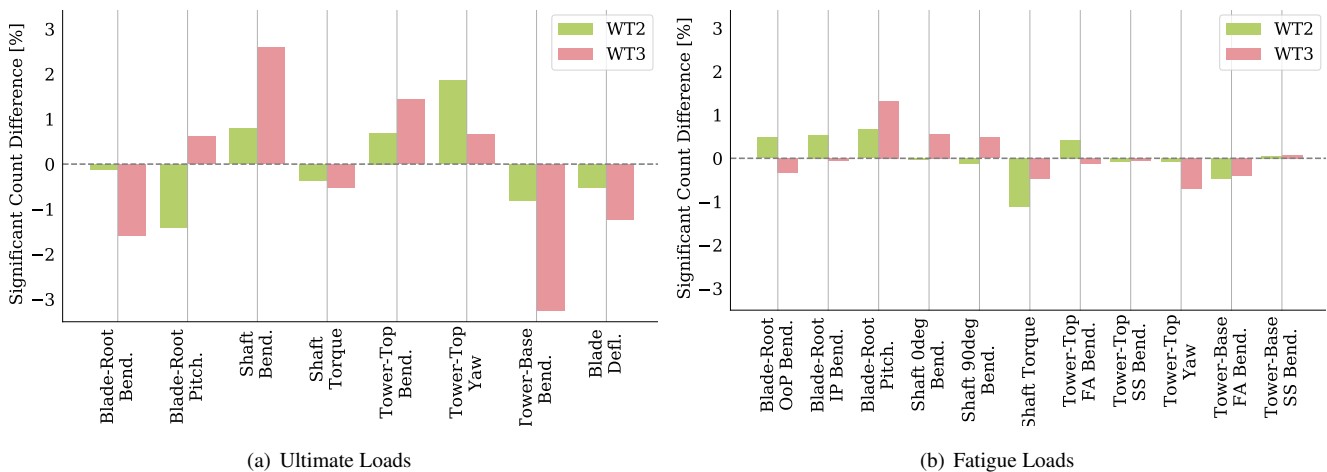

(a) Ultimate Loads        (b) Fatigue Loads

**Figure 8.** The percent difference in significant event counts for WT2 and WT3 relative to WT1, based on QoI.

To further investigate which QoI were influenced by the input parameters, Figure 6 shows tabulated results for the number
255 of ultimate load significant events for each input parameter, separated by QoI. Figure 6(a) shows the absolute value, while
Figures 6(b) and 6(c) show the difference from WT1 results for the waked turbines. Similar results are shown in Figure 7 for
fatigue loads. For all turbines, the top three QoIs that contribute to load sensitivity were tower-top bending, tower-top yaw
moment, and low-speed shaft bending, though the exact ranking was different for all turbines. For each turbine, 14-18% of
the significant events resulting from these load channels. The frequency with which QoIs triggered significant events differs,
260 as summarized in Figure 8, which shows the percent difference in significant event counts for WT2 and WT3 relative to WT1
as calculated by Equation 5, but based on QoI instead of input parameter. For WT2, the most differences occur for blade-root
pitching moment, reduced by 1.5% and tower-top yaw moment, increased by 1.8%. For WT3, the most differences occur for
tower-base bending moment, reduced by 3.3%, and shaft bending moment, increased by 2.6%. Similar results were seen for
fatigue results, though to a lesser extent.

265 When looking only at the contribution of tertiary parameters in Figures 6(b) and 6(c), blade-root pitching moment stands
out the most for all turbines, though nearly twice as much for WT3 compared to WT1. Overall, WT3 loads were up to $8\times$
more sensitive to tertiary parameter variation as compared to WT1, with this highest increase occurring for low-speed shaft
bending ultimate loads. Tower-top bending, tower-top yaw moment, and low-speed shaft bending contributed the most to load
sensitivity for all turbines. Though the top QoIs were the same, the exact ranking and amount of events differed.

270 Figure 7 shows tabulated results for the number of fatigue load significant events for each input parameter, separated by
QoI. For all turbines, the top three QoIs that contribute to load sensitivity were blade-root in-plane bending, low-speed shaft
$0°$ bending, and low-speed shaft $90°$ bending. For each turbine, 26-28% of the significant events resulting from these load
channels. The QoI that were most sensitive for WT1 were in-plane blade-root moment and low-speed shaft bending. For
waked turbines, the most sensitive QoIs were in-plane blade-root bending moment, inflow TI, and $0°$ low-speed shaft bending.





**Table 6.** Tabulated results for the number of ultimate load significant events for each input parameter, separated by QoI. Cells are colored by the count value, with darker blue representing a more positive count and darker red representing a more negative count.

(a) WT1

| | $\alpha$ | $\beta$ | $\sigma_u$ | $\sigma_v$ | $\sigma_w$ | $L_u$ | $L_v$ | $L_w$ | $a_u$ | $a_v$ | $a_w$ | $b_u$ | $b_v$ | $b_w$ | $\gamma$ | $PV_{uw}$ | $PV_{uv}$ | $PV_{vw}$ | WD | $\rho$ | $\Theta_{T1}$ | $\Theta_{T2}$ | $\Theta_{T3}$ | $C_{NW}$ | $k_{vAmb}$ | $k_{vShr}$ | $C_M$ | $f_c$ |
|---|---|---|---|---|---|---|---|---|---|---|---|---|---|---|---|---|---|---|---|---|---|---|---|---|---|---|---|---|
| Blade-Root Bend. | 26 | 1 | 28 | 3 | 1 | 0 | 3 | 1 | 13 | 1 | 0 | 9 | 0 | 0 | 5 | 1 | 1 | 0 | 0 | 0 | 7 | 0 | 0 | 0 | 0 | 0 | 0 | 0 |
| Blade-Root Pitch. | 17 | 2 | 27 | 2 | 0 | 1 | 4 | 3 | 7 | 3 | 3 | 5 | 0 | 1 | 5 | 1 | 0 | 1 | 0 | 2 | 1 | 0 | 0 | 0 | 0 | 0 | 0 | 0 |
| Shaft Bend. | 49 | 0 | 36 | 0 | 0 | 2 | 1 | 2 | 8 | 0 | 0 | 9 | 0 | 1 | 3 | 2 | 0 | 0 | 0 | 0 | 3 | 0 | 0 | 0 | 0 | 0 | 0 | 0 |
| Shaft Torq. | 14 | 0 | 41 | 0 | 1 | 1 | 1 | 0 | 6 | 0 | 0 | 9 | 0 | 0 | 3 | 0 | 0 | 0 | 0 | 0 | 1 | 0 | 0 | 0 | 0 | 0 | 0 | 0 |
| Tower-Top Bend. | 56 | 0 | 33 | 0 | 0 | 3 | 1 | 2 | 9 | 0 | 1 | 8 | 2 | 0 | 1 | 1 | 0 | 0 | 0 | 0 | 3 | 0 | 0 | 0 | 0 | 0 | 0 | 0 |
| Tower-Top Yaw | 20 | 0 | 54 | 2 | 0 | 1 | 1 | 1 | 14 | 0 | 1 | 7 | 0 | 0 | 5 | 0 | 1 | 0 | 0 | 0 | 4 | 0 | 0 | 0 | 0 | 0 | 0 | 0 |
| Tower-Base Bend. | 13 | 1 | 14 | 7 | 1 | 8 | 3 | 1 | 10 | 1 | 1 | 8 | 1 | 0 | 4 | 0 | 0 | 1 | 0 | 0 | 11 | 0 | 0 | 0 | 0 | 0 | 0 | 0 |
| Blade Defl. | 33 | 0 | 17 | 2 | 1 | 0 | 3 | 0 | 12 | 2 | 0 | 10 | 0 | 0 | 2 | 1 | 0 | 0 | 0 | 0 | 10 | 0 | 0 | 0 | 0 | 0 | 0 | 0 |
| Generator Power | 14 | 0 | 3 | 0 | 1 | 1 | 1 | 0 | 3 | 1 | 0 | 3 | 1 | 0 | 2 | 1 | 0 | 0 | 0 | 0 | 2 | 0 | 0 | 0 | 0 | 0 | 0 | 0 |
| Farm Gen. Power | 18 | 0 | 8 | 0 | 0 | 0 | 0 | 0 | 3 | 1 | 0 | 6 | 0 | 0 | 2 | 1 | 0 | 2 | 0 | 0 | 1 | 0 | 0 | 0 | 0 | 0 | 0 | 0 |

(b) WT2

| | $\alpha$ | $\beta$ | $\sigma_u$ | $\sigma_v$ | $\sigma_w$ | $L_u$ | $L_v$ | $L_w$ | $a_u$ | $a_v$ | $a_w$ | $b_u$ | $b_v$ | $b_w$ | $\gamma$ | $PV_{uw}$ | $PV_{uv}$ | $PV_{vw}$ | WD | $\rho$ | $\Theta_{T1}$ | $\Theta_{T2}$ | $\Theta_{T3}$ | $C_{NW}$ | $k_{vAmb}$ | $k_{vShr}$ | $C_M$ | $f_c$ |
|---|---|---|---|---|---|---|---|---|---|---|---|---|---|---|---|---|---|---|---|---|---|---|---|---|---|---|---|---|
| Blade-Root Bend. | 5 | -1 | 3 | -1 | -1 | 0 | -2 | 0 | -1 | -1 | 0 | -1 | 0 | 0 | -2 | 1 | -1 | 0 | 0 | 0 | -7 | 4 | 0 | 0 | 0 | 0 | 0 | 0 |
| Blade-Root Pitch. | -2 | -2 | -4 | 1 | 0 | 3 | -4 | -2 | -3 | -3 | -2 | 3 | 1 | -1 | -3 | 1 | 1 | 0 | 0 | -1 | -1 | 3 | 0 | 0 | 0 | 0 | 0 | 0 |
| Shaft Bend. | 1 | 0 | 4 | 1 | 0 | 0 | 0 | -2 | -1 | 0 | 1 | 1 | 0 | -1 | 0 | 0 | 0 | 0 | 0 | 0 | -3 | 1 | 0 | 0 | 0 | 0 | 0 | 0 |
| Shaft Torq. | 0 | 1 | -3 | 0 | -1 | -1 | -1 | 0 | 0 | 0 | 0 | -2 | 0 | 0 | 1 | 1 | 0 | 0 | 0 | 0 | -1 | 0 | 0 | 0 | 0 | 0 | 0 | 0 |
| Tower-Top Bend. | 4 | 0 | 3 | 1 | 0 | -1 | -1 | -1 | -3 | 1 | 0 | 0 | -2 | 0 | 1 | 1 | 0 | 0 | 0 | 0 | -3 | 1 | 0 | 0 | 0 | 0 | 0 | 0 |
| Tower-Top Yaw | 6 | 0 | 0 | 0 | 1 | 1 | 1 | 0 | -1 | 0 | 0 | 2 | 0 | 0 | 3 | 1 | -1 | 0 | 0 | 0 | -4 | 2 | 0 | 0 | 0 | 0 | 0 | 0 |
| Tower-Base Bend. | -2 | -1 | 2 | -2 | -1 | -2 | 2 | -1 | 3 | 0 | -1 | -2 | -1 | 0 | -1 | 3 | 0 | -1 | 0 | 0 | -11 | 6 | 0 | 0 | 0 | 0 | 0 | 0 |
| Blade Defl. | -1 | 0 | -1 | -1 | -1 | 1 | -2 | 1 | -3 | -1 | 1 | -2 | 1 | 1 | 0 | 2 | 0 | 0 | 0 | 0 | -10 | 7 | 0 | 0 | 0 | 0 | 0 | 0 |
| Generator Power | 1 | 0 | 0 | 1 | 0 | -1 | -1 | 0 | -1 | -1 | 0 | 1 | 0 | 0 | -1 | 0 | 0 | 0 | 0 | 0 | -2 | 0 | 0 | 0 | 0 | 0 | 0 | 0 |

(c) WT3

| | $\alpha$ | $\beta$ | $\sigma_u$ | $\sigma_v$ | $\sigma_w$ | $L_u$ | $L_v$ | $L_w$ | $a_u$ | $a_v$ | $a_w$ | $b_u$ | $b_v$ | $b_w$ | $\gamma$ | $PV_{uw}$ | $PV_{uv}$ | $PV_{vw}$ | WD | $\rho$ | $\Theta_{T1}$ | $\Theta_{T2}$ | $\Theta_{T3}$ | $C_{NW}$ | $k_{vAmb}$ | $k_{vShr}$ | $C_M$ | $f_c$ |
|---|---|---|---|---|---|---|---|---|---|---|---|---|---|---|---|---|---|---|---|---|---|---|---|---|---|---|---|---|
| Blade-Root Bend. | -4 | 0 | 3 | -1 | 0 | 3 | -3 | -1 | -3 | -1 | 0 | -3 | 0 | 0 | -1 | -1 | 0 | 0 | 1 | 0 | -7 | 1 | 4 | 0 | 0 | 0 | 1 | 0 |
| Blade-Root Pitch. | 1 | 1 | -2 | 0 | 1 | 4 | -2 | -2 | -1 | 0 | 1 | -2 | 1 | -1 | 2 | 2 | 3 | 0 | 1 | -1 | -1 | 0 | 1 | 0 | 0 | 0 | 1 | 0 |
| Shaft Bend. | 1 | 1 | 1 | 0 | 1 | 0 | 0 | 0 | 2 | 1 | 3 | -1 | 1 | 0 | 3 | -1 | 3 | 2 | 1 | 1 | -2 | 1 | 3 | 0 | 1 | 1 | 1 | 1 |
| Shaft Torq. | -2 | 1 | -5 | 1 | -1 | -1 | -1 | 0 | 1 | 0 | 0 | -2 | 0 | 0 | 1 | 0 | 0 | 0 | 1 | 0 | 0 | 1 | 1 | 0 | 1 | 0 | 0 | 1 |
| Tower-Top Bend. | -4 | 1 | -1 | 1 | 1 | -1 | 0 | 0 | -1 | 2 | 1 | 0 | 0 | 1 | 3 | 2 | 2 | 1 | 1 | 1 | -2 | 1 | 2 | 0 | 1 | 1 | 1 | 1 |
| Tower-Top Yaw | 4 | 0 | -2 | -1 | 0 | 2 | -1 | 0 | -2 | 0 | 1 | 1 | 1 | 0 | 2 | 2 | 0 | 2 | 0 | 0 | -4 | 0 | 3 | 0 | 0 | 0 | 0 | 0 |
| Tower-Base Bend. | -4 | -1 | 5 | -7 | -1 | -3 | -2 | -1 | 2 | -1 | -1 | -1 | 0 | 0 | -2 | 0 | 0 | -1 | 0 | 0 | -11 | 0 | 2 | 0 | 0 | 0 | 0 | 0 |
| Blade Defl. | 1 | 1 | 0 | -2 | -1 | 2 | -2 | 0 | -1 | -1 | 0 | 0 | 0 | 0 | 1 | -1 | 1 | 0 | 0 | 0 | -10 | 0 | 3 | 0 | 0 | 0 | 0 | 0 |
| Generator Power | 3 | 0 | 1 | 1 | 1 | -1 | 0 | 1 | 0 | 1 | 1 | 0 | 0 | 1 | 0 | 0 | 1 | 0 | 1 | 0 | -1 | 0 | 2 | 0 | 0 | 0 | 1 | 0 |

For WT3, inflow TI was the most sensitive QoI. When considering Figure 8(b), the highest increase in QoI sensitivity were for blade-root pitching, with WT3 resulting in $2.5\times$ as many significant events from tertiary parameters compared to WT1.





**Table 7.** Tabulated results for the number of fatigue load significant events for each input parameter, separated by QoI. Cells are colored by the count value, with darker blue representing a more positive count and darker red representing a more negative count.

(a) WT1

| | $\alpha$ | $\beta$ | $\sigma_u$ | $\sigma_v$ | $\sigma_w$ | $L_u$ | $L_v$ | $L_w$ | $a_u$ | $a_v$ | $a_w$ | $b_u$ | $b_v$ | $b_w$ | $\gamma$ | $PV_{uw}$ | $PV_{uv}$ | $PV_{vw}$ | WD | $\rho$ | $\Theta_{T1}$ | $\Theta_{T2}$ | $\Theta_{T3}$ | $C_{NW}$ | $k_{vAmb}$ | $k_{vShr}$ | $C_M$ | $f_c$ |
|---|---|---|---|---|---|---|---|---|---|---|---|---|---|---|---|---|---|---|---|---|---|---|---|---|---|---|---|---|
| Blade-Root OoP | 18 | 1 | 42 | 2 | 1 | 3 | 1 | 2 | 10 | 0 | 0 | 13 | 2 | 0 | 10 | 1 | 1 | 0 | 0 | 1 | 0 | 0 | 0 | 0 | 0 | 0 | 0 | 0 |
| Blade-Root IP Bend. | 57 | 1 | 42 | 3 | 0 | 1 | 0 | 0 | 9 | 0 | 0 | 10 | 0 | 0 | 6 | 0 | 0 | 0 | 0 | 0 | 8 | 0 | 0 | 0 | 0 | 0 | 0 | 0 |
| Blade-Root Pitch. | 15 | 3 | 36 | 4 | 1 | 2 | 2 | 1 | 6 | 1 | 1 | 7 | 2 | 1 | 8 | 0 | 2 | 1 | 0 | 0 | 2 | 0 | 0 | 0 | 0 | 0 | 0 | 0 |
| Shaft 0° Bend. | 61 | 2 | 38 | 1 | 0 | 0 | 0 | 0 | 9 | 0 | 0 | 8 | 0 | 0 | 3 | 0 | 0 | 0 | 0 | 0 | 5 | 0 | 0 | 0 | 0 | 0 | 0 | 0 |
| Shaft 90° Bend. | 61 | 1 | 40 | 1 | 0 | 1 | 0 | 0 | 8 | 0 | 0 | 6 | 0 | 0 | 4 | 0 | 0 | 0 | 0 | 0 | 4 | 0 | 0 | 0 | 0 | 0 | 0 | 0 |
| Shaft Torq. | 8 | 0 | 66 | 1 | 2 | 1 | 1 | 1 | 7 | 1 | 0 | 11 | 1 | 2 | 8 | 0 | 2 | 1 | 0 | 0 | 1 | 0 | 0 | 0 | 0 | 0 | 0 | 0 |
| Tower-Top FA Bend. | 4 | 1 | 55 | 0 | 0 | 1 | 0 | 0 | 2 | 1 | 0 | 3 | 0 | 0 | 3 | 0 | 0 | 0 | 0 | 0 | 41 | 0 | 0 | 0 | 0 | 0 | 0 | 0 |
| Tower-Top SS Bend. | 7 | 1 | 59 | 2 | 0 | 4 | 1 | 2 | 6 | 1 | 0 | 8 | 0 | 1 | 13 | 0 | 0 | 0 | 0 | 0 | 1 | 0 | 0 | 0 | 0 | 0 | 0 | 0 |
| Tower-Top Yaw | 6 | 2 | 63 | 1 | 1 | 4 | 1 | 1 | 5 | 0 | 0 | 9 | 1 | 0 | 7 | 1 | 1 | 0 | 0 | 0 | 4 | 0 | 0 | 0 | 0 | 0 | 0 | 0 |
| Tower-Base FA Bend. | 3 | 0 | 48 | 7 | 0 | 9 | 9 | 1 | 7 | 5 | 0 | 1 | 1 | 1 | 9 | 1 | 1 | 1 | 0 | 1 | 11 | 0 | 0 | 0 | 0 | 0 | 0 | 0 |
| Tower-Base SS Bend. | 4 | 0 | 51 | 4 | 0 | 7 | 4 | 0 | 12 | 2 | 0 | 3 | 0 | 0 | 10 | 0 | 0 | 0 | 0 | 1 | 8 | 0 | 0 | 0 | 0 | 0 | 0 | 0 |
| Generator Power | 11 | 3 | 12 | 5 | 0 | 3 | 3 | 1 | 11 | 2 | 1 | 17 | 0 | 0 | 9 | 0 | 0 | 0 | 0 | 0 | 20 | 0 | 0 | 0 | 0 | 0 | 0 | 0 |
| Turbulence Intensity | 1 | 1 | 87 | 3 | 1 | 3 | 2 | 2 | 2 | 0 | 1 | 2 | 3 | 1 | 0 | 4 | 3 | 2 | 0 | 0 | 0 | 0 | 0 | 0 | 0 | 0 | 0 | 0 |

(b) WT2

| | $\alpha$ | $\beta$ | $\sigma_u$ | $\sigma_v$ | $\sigma_w$ | $L_u$ | $L_v$ | $L_w$ | $a_u$ | $a_v$ | $a_w$ | $b_u$ | $b_v$ | $b_w$ | $\gamma$ | $PV_{uw}$ | $PV_{uv}$ | $PV_{vw}$ | WD | $\rho$ | $\Theta_{T1}$ | $\Theta_{T2}$ | $\Theta_{T3}$ | $C_{NW}$ | $k_{vAmb}$ | $k_{vShr}$ | $C_M$ | $f_c$ |
|---|---|---|---|---|---|---|---|---|---|---|---|---|---|---|---|---|---|---|---|---|---|---|---|---|---|---|---|---|
| Blade-Root OoP | -3 | 1 | 1 | -2 | 0 | -1 | 3 | 0 | -1 | 1 | 1 | -2 | -1 | 2 | 1 | 1 | 1 | 1 | 0 | 0 | 0 | 1 | 0 | 0 | 0 | 0 | 0 | 0 |
| Blade-Root IP Bend. | 4 | 1 | -2 | -2 | 0 | 0 | 1 | 0 | 1 | 1 | 1 | 0 | 0 | 1 | 2 | 0 | 0 | 0 | 0 | 0 | -8 | 4 | 0 | 0 | 0 | 0 | 0 | 0 |
| Blade-Root Pitch. | 0 | -3 | 2 | 0 | 0 | 0 | 0 | -1 | 1 | -1 | 2 | 0 | 0 | 0 | 0 | 2 | 0 | 0 | 1 | 1 | -1 | 2 | 0 | 1 | 0 | 0 | 0 | 1 |
| Shaft 0° Bend. | 1 | -1 | -2 | -1 | 0 | 0 | 0 | 1 | 0 | 1 | 0 | -2 | 0 | 1 | 0 | 0 | 0 | 0 | 0 | 0 | -5 | 3 | 0 | 0 | 0 | 0 | 0 | 0 |
| Shaft 90° Bend. | 0 | 0 | -5 | -1 | 0 | -1 | 0 | 1 | 1 | 1 | 0 | 0 | 0 | 1 | 0 | 0 | 0 | 0 | 0 | 0 | -4 | 2 | 0 | 0 | 0 | 0 | 0 | 0 |
| Shaft Torq. | -3 | 0 | -9 | -1 | -2 | -1 | 0 | -1 | 0 | -1 | 0 | 2 | 1 | -2 | 1 | 0 | -2 | -1 | 0 | 0 | -1 | 1 | 0 | 0 | 0 | 0 | 0 | 0 |
| Tower-Top FA Bend. | 0 | 0 | -3 | 1 | 0 | -1 | 0 | 0 | 2 | 0 | 0 | 0 | 1 | 1 | 3 | 1 | 0 | 0 | 0 | 0 | -41 | 39 | 0 | 0 | 0 | 0 | 0 | 0 |
| Tower-Top SS Bend. | -1 | 0 | -2 | 1 | 0 | -2 | 0 | 0 | 1 | 0 | 0 | 1 | 0 | -1 | -1 | 0 | 0 | 0 | 0 | 0 | -1 | 1 | 0 | 0 | 0 | 0 | 0 | 0 |
| Tower-Top Yaw | 1 | -1 | -4 | 1 | -1 | 1 | -1 | 0 | 0 | 1 | 1 | -1 | 0 | 1 | 1 | -1 | -1 | 0 | 0 | 0 | -4 | 3 | 0 | 0 | 0 | 0 | 0 | 0 |
| Tower-Base FA Bend. | -2 | 0 | -1 | -1 | 0 | -3 | -2 | 1 | 1 | -2 | 0 | -1 | 1 | -1 | 0 | 1 | 0 | -1 | 0 | -1 | -11 | 12 | 0 | 0 | 0 | 0 | 0 | 0 |
| Tower-Base SS Bend. | -2 | 0 | -1 | 0 | 0 | -1 | -1 | 1 | 4 | -1 | 0 | 1 | 0 | 0 | 2 | 0 | 0 | 0 | 0 | -1 | -8 | 5 | 0 | 0 | 0 | 0 | 0 | 0 |
| Generator Power | -5 | -3 | 1 | 0 | 0 | 0 | -2 | -1 | 0 | -1 | 0 | -2 | 0 | 0 | -5 | 0 | 0 | 0 | 1 | 0 | -20 | 19 | 0 | 0 | 0 | 0 | 0 | 0 |
| Turbulence Intensity | 0 | 1 | -15 | 0 | 1 | 0 | -1 | 0 | 0 | 0 | 0 | 1 | 2 | 3 | 5 | 1 | -1 | 1 | 8 | 1 | 1 | 0 | 0 | 1 | 0 | 0 | 0 | 0 |

(c) WT3

| | $\alpha$ | $\beta$ | $\sigma_u$ | $\sigma_v$ | $\sigma_w$ | $L_u$ | $L_v$ | $L_w$ | $a_u$ | $a_v$ | $a_w$ | $b_u$ | $b_v$ | $b_w$ | $\gamma$ | $PV_{uw}$ | $PV_{uv}$ | $PV_{vw}$ | WD | $\rho$ | $\Theta_{T1}$ | $\Theta_{T2}$ | $\Theta_{T3}$ | $C_{NW}$ | $k_{vAmb}$ | $k_{vShr}$ | $C_M$ | $f_c$ |
|---|---|---|---|---|---|---|---|---|---|---|---|---|---|---|---|---|---|---|---|---|---|---|---|---|---|---|---|---|
| Blade-Root OoP | -2 | 0 | -1 | -1 | 0 | -2 | 1 | -1 | 0 | 0 | 1 | -1 | -1 | 0 | 1 | -1 | -1 | 0 | 0 | -1 | 0 | 0 | 0 | 0 | 0 | 0 | 0 | 0 |
| Blade-Root IP Bend. | 0 | 0 | -3 | -2 | 0 | 0 | 1 | 0 | 0 | 1 | 0 | 1 | 0 | 1 | 1 | 0 | 0 | 0 | 0 | 0 | -8 | 0 | 2 | 0 | 0 | 0 | 0 | 0 |
| Blade-Root Pitch. | -4 | -1 | 2 | -2 | 2 | 2 | 0 | 2 | 1 | 0 | 1 | 2 | -1 | 1 | -2 | 2 | 1 | 0 | 1 | 1 | -1 | 1 | 2 | 1 | 1 | 1 | 1 | 1 |
| Shaft 0° Bend. | 3 | -1 | 0 | -1 | 0 | 0 | 0 | 1 | 0 | 1 | 0 | 1 | 0 | 1 | 1 | 0 | 0 | 0 | 0 | 0 | -5 | 0 | 2 | 0 | 0 | 0 | 0 | 0 |
| Shaft 90° Bend. | 3 | 0 | -3 | -1 | 0 | -1 | 0 | 0 | 1 | 0 | 0 | 4 | 0 | 1 | 1 | 0 | 0 | 0 | 0 | 0 | -4 | 0 | 1 | 0 | 0 | 0 | 0 | 0 |
| Shaft Torq. | -1 | 0 | -7 | 0 | -2 | 0 | -1 | 0 | 4 | -1 | 1 | 0 | 0 | -2 | -1 | 0 | -1 | 0 | 1 | 0 | -1 | 0 | 0 | 0 | 0 | 0 | 0 | 0 |
| Tower-Top FA Bend. | -2 | -1 | -3 | 1 | 0 | -1 | 0 | 0 | 0 | -1 | 0 | 2 | 0 | 0 | 3 | 0 | 0 | 0 | 0 | 0 | -41 | 0 | 37 | 0 | 0 | 0 | 0 | 0 |
| Tower-Top SS Bend. | -2 | 0 | -1 | 1 | 0 | -3 | 0 | 0 | 1 | 0 | 0 | 1 | 0 | 0 | -2 | 0 | 0 | 0 | 0 | 0 | -1 | 0 | 1 | 0 | 0 | 0 | 0 | 0 |
| Tower-Top Yaw | -1 | -1 | -3 | 0 | -1 | -1 | -1 | -1 | 0 | 0 | 0 | -2 | -1 | 1 | 1 | -1 | -1 | 0 | 1 | 0 | -4 | 0 | 1 | 0 | 0 | 0 | 0 | 0 |
| Tower-Base FA Bend. | -2 | 1 | -5 | 1 | 2 | -2 | -3 | 2 | 2 | -2 | 1 | 0 | -1 | 0 | -2 | 1 | -1 | 2 | 0 | -1 | -11 | 0 | 8 | 0 | 0 | 0 | 0 | 0 |
| Tower-Base SS Bend. | 0 | 0 | 4 | -1 | 0 | -1 | -3 | 0 | 2 | -2 | 0 | 4 | 0 | 0 | -2 | 0 | 0 | 0 | 1 | -1 | -8 | 0 | 4 | 0 | 0 | 0 | 0 | 0 |
| Generator Power | -7 | -3 | 1 | -3 | 2 | -1 | -2 | -1 | -1 | -1 | -1 | -4 | 0 | 0 | -7 | 0 | 1 | 1 | 0 | 0 | -20 | 0 | 12 | 0 | 0 | 0 | 0 | 0 |
| Turbulence Intensity | 0 | 3 | -16 | 1 | 3 | -1 | 1 | 0 | 3 | 2 | 2 | 2 | 0 | 2 | 5 | 0 | -1 | 0 | 15 | 0 | 0 | 0 | 0 | 0 | 0 | 1 | 0 | 1 | 1 |



# 5   Conclusions

This work aimed to highlight the relative importance of inflow and wake parameters for fatigue and ultimate load sensitivity. This was accomplished by developing metrics to assess the sensitivity of several turbine load measurements, and assessing how this sensitivity changes with varying inflow and wake conditions. The sensitivity was assessed using an EE method, considering a wide range of possible wind inflow and wake conditions. From these sensitivity values, a thresholding method was used to determine when a sensitivity value was classified as a "significant event". From this, the number of "significant events" triggered by varying each parameter was analyzed, along with which aeroelastic QoI were most effected. The results from this work can be used to better inform not only the turbine design process and site-suitability analyses, but also help identify important measurement quantities when designing wind farm experiments.

The results of this work show that for both waked and non-waked turbines, ambient turbulence in the primary wind direction and shear were the most sensitive parameters for turbine fatigue and ultimate loads. Secondary parameters of importance for all turbines were identified as yaw misalignment, u-direction integral length, and u components of the IEC coherence model, as well as the exponent. The tertiary parameters of performance differ between waked and non-waked turbines. Tertiary effects for ultimate loads of waked turbines were veer; non-streamwise components of the IEC coherence model; Reynolds stresses; wind direction; air density; and several wake calibration parameters, with these tertiary effects accounting for up to $9.0\%$ of the significant events for waked turbines. For fatigue loads, the tertiary effects of waked turbines were the vertical turbulence standard deviation; lateral and vertical wind integral lengths; lateral and vertical wind components of the IEC coherence model; Reynolds stresses; wind direction; and all wake calibration parameters, with these tertiary effects accounting for up to $5.4\%$ of the significant events of waked turbines. This information shows the increased importance of non-streamwise wind components and wake parameters in fatigue and ultimate load sensitivity of downstream turbines. Additionally, the most effected QoIs differed between waked and unwaked turbines.

*Author contributions.*  KS led the investigation. ANR developed the EE methodology approach used in the parameter study. JJ provided the conceptualization and supervision for this project. KS prepared the article, with support from ANR and JJ.

*Competing interests.*  The authors declare that they have no conflict of interest.

*Disclaimer.*  The views expressed in the article do not necessarily represent the views of the DOE or the U.S. Government

*Acknowledgements.*  This work was authored by Alliance for Sustainable Energy, LLC, the manager and operator of the National Renewable Energy Laboratory for the U.S. Department of Energy (DOE) under Contract No. DE-AC36-08GO28308. Funding provided by Department



of Energy Office of Energy Efficiency and Renewable Energy, Wind Energy Technologies Office. The views expressed in the article do
not necessarily represent the views of the DOE or the U.S. Government. The U.S. Government retains and the publisher, by accepting the
article for publication, acknowledges that the U.S. Government retains a nonexclusive, paid-up, irrevocable, worldwide license to publish or
reproduce the published form of this work, or allow others to do so, for U.S. Government purposes.

The research was performed using computational resources sponsored by the Department of Energy's Office of Energy Efficiency and
Renewable Energy and located at the National Renewable Energy Laboratory.





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
