# Peer review of "Sensitivity Analysis of Turbine Fatigue and Ultimate Loads to Wind and Wake Characteristics in a Small Wind Farm"

_Wind Energy Science, 2021_

## Author Response (AR1)

**Reviewer Responses for**

**Sensitivity Analysis of Turbine Fatigue and Ultimate Loads to Wind and Wake Characteristics in a Small Wind Farm**

**Reviewer #1**

Investigation into how different environmental conditions effect turbine loads is very important for a number of perspectives within Wind Energy. So, the aim of this work is very compelling. However, it was difficult for me to judge the significance of this work for a number of reasons. **First, the literature review is totally inadequate so it fails to adequately place itself in the wider body of literature**. **The focus of the paper is on wake effects, however, the wake model is not well described. Furthermore, due to the lack of review it's also difficult to assess what assumptions were used in the modelling**. **Furthermore, there is no validation of the chosen model.** So it's also difficult to determine how well this analysis reflects reality. **In the presentation of the results, the authors tend to show too much data in the plots, which makes the plots difficult to read and understand.** I would encourage the authors to focus on the important parameters and group the rest ins "misc." or "other**". Furthermore, in the discussion, the authors merely highlight the results given in the tables. I think that the discussion can be greatly enhanced if they could give more insights and explaination on how or why these ressults came about. To what extent are these conclusions a reflection of reality or maybe effected by different choices in the modelling and analysis?** Otherwise, the article reads too much like a technical report. Detailed comments are given below.

1. The abstract is too long and verbose. It is written more as an introduction. Please make it more concise.

   **Author Response:** The authors have shortened the abstract some but believe that all remaining information is necessary.

2. The abstract also fails to mention the model that was used to calculate the loads, I assume FAST. Additionally, the author uses highly technical language to describe other models (e.g. IEC Kaimal turbulence spectrum with IEC exponential coherence model), it would make the work more readable to a larger audience if they used more coloquial terms like Turbsim.

   **Author Response:** The authors have added references to OpenFAST and TurbSim to the abstract. The specific information about the turbulence model remains because there are several options available within TurbSim.

3. In the abstract and introduction, the author only mentions their own previous work on the subject and claims that their work is unique. However, they fail to mention any other work for example "https://wes.copernicus.org/articles/3/767/2018/". On the subject of loads in general, the impact of wakes on turbines, the sensitivity of turbulence and inflow on loads etc. there are multiple research groups that are looking into these things (DTU, University of Oldenburg, TUM, etc.). There is a huge body of work already and failing to give a proper review of the relevant literature makes it difficult to judge where this works fits within the

larger body of literature. The authors need to give a more comprehensive review of the state of the art on this subject.

**Author Response:** The authors have added a paragraph to the introduction that focuses on past sensitivity analysis work as it relates to wind farms. This should provide sufficient justification into the novelty of this work.

4. Author Response: Further to the review of literature, there are actually multiple approaches for evaluating sensitivity. Sobol indices for example. Each of these methods look at different aspectcs (local sensitivity, variance based, etc.). It would be helpful if the author could place their choice amongst these competing methods and explain why they chose their method and how that choice may alter the conclusions.

**Author Response:** These points were covered in a previous publication by the authors, https://doi.org/10.5194/wes-4-479-2019. This have been reinforced in the present publication, and readers are more heavily encouraged to see that publication for that information.

5. The author uses the term "waked" and "unwaked", these are proper english words. However, they are misused by the author when they are used to differentiate between operating within a wake or not. According to the dictionary, waked refers to either the state of being awake or to an event that occurs after a funeral. To avoid confusion, it's important that the author uses the correct terminology.

**Author Response:** This is accepted terminology within the wind energy field. The authors feel it is appropriate to keep this language, and using a less-common term would add confusion.

6. Line 58 when you define QoI, capitolize Quantities of Interest.

**Author Response:** According to our internal communications team, it is preferred to not capitalize these words.

7. In the methods section the author mentions the use of established tools. Regardless it would still help to give citations where people can learn about the details of those models, (Elastodyn, ServoDyn, etc.).

**Author Response:** References have also been added to OpenFAST, ElastoDyn, and ServoDyn.

8. The title "Case Study" for section 2.2 is too vague and implies that some results will be given.

**Author Response:** The word "Description" has been added to this section title

9. I am not sure that section 3 needs it's own section. Section 3 is a method and I think would fit better as a sub section in section 2.

**Author Response:** We did this to separate the established methods used for this work, such as FAST.Farm and OpenFAST, with the work that was studies in the paper, such as the EE method and what parameters were varied.

10. Overall it is unclear how wake effects were modelled. There are little details on the model, other than a description of various parameters. Since this is the main focus of the paper, it is very important that a better description of this model is given. The other thing that I am concerned with, is that it's not clear how well the chosen wake model fits with reality. It would be nice to see some validation of the wake model with either high fidelity tools or with experimental data. Maybe previous work can be leaned on. A parameter study on a model is rather useless if one does not know how well the model itself describes reality. Again, a literature review on different wake modelling would also be helpful here too. What fidelity level is this work based on? What assumptions are made in the modelling? I am also curious whether this model is based on partial wake coverage or not.

    **Author Response:** Wake modeling was done with FAST.Farm, which has been heavily documented and validated. Further details have been added to this section, including past validation studies of the model.

11. Also, another interesting parameter that I think would be important for a study like this is the distance between turbines.

    **Author Response:** The authors agree that this would be an interesting aspect to explore. However, given that this study already involved 130,500 FAST.Farm simulations and exploring turbine spacing would require more simulations, this would be more appropriate for a separate study. Also, turbine spacing is different from other parameters used in that turbine spacing is generally a known quantity for a given wind farm application whereas this study considers parameters that have more uncertainty/variability.

12. There are a lot of parameters discussed in the paper, in tables and figures they are represented as symbols that have been defined in the text. Given the large number of parameters and the fact that the parameters are the focus of the study, it would be helpful of a nomenclature was given to list the definitions of all the symbols.

    **Author Response:** The authors agree, but this is not a part of the journal format. The input parameter symbols are all listed together in Table 2, which the authors believe is sufficient.

13. The text in figure 2 is too small and difficult to read. Also, the colored text for inflow parameters is difficult to see with the hazy background.

    **Author Response:** We have increased the font size in this plot, and added a background to the text boxes to further improve readability.

14. Figure 4 shows a lot of information... It's a funny way to show things, but I can understand that they are trying to focus in on the significant results. However, because there is so much data, it's difficult to comrehend the plots. I would reccomend only populating the figures with results that exceed the threshold and try to give better labels. It's really confusing, because

the yellow curves almost completely obscure the other colors. The legend is only for the blue data. It's clear that they run out of symbols to differentiate the different data. So, because they are trying to show all data, it's actually becomes too confusing to understand these plots.

**Author Response:** This figure is used primarily to demonstrate the method by which significant event identification was done. It is intended more for demonstration and explaining the process, with the plots further down being used to summarize and pull out important information.

The colors are used to distinguish between different turbines. The same symbols are used for each turbine. This is not included in the legend because it would be redundant and take up too much space. However, this point has been clarified in the figure caption. Additionally, unique symbols were only used for parameters that primarily contributed to the significant events count. This was done intentionally to draw more attention to them.

15. Could figure 6 and 7 be plotted in log scale for the y axis? Again, it appears that a lot of data is given ... the clarity could be improved by focusing on the important stuff. I really think that you need to have a category "misc." or "other" to group together the data that is not important.

**Author Response:** The authors intentionally did not use a log scale because visually this would artificially make certain count numbers appear more substantial than others. Therefore, a linear scale was used and the range was limited to focus on the tertiary effects.

16. Table 6 is not described well enough for me to understand the data. Why is WT1 all blue, while 2 and 3 are a mix of red and blue? I suppose 2 and 3 are relative to 1, but that is not mentioned in the caption. Also, instead of giving raw counts, it would be more helpful to normalize the data in percentages. Otherwise, it's difficult to judge the significance of these counts.

**Author Response:** The description of this table has been clarified in the caption and in the text.

In terms of presenting the data as percentages, the authors had previously investigated this method. However, because the total number of counts varies between QoI and parameter, this resulted in some quantities showing up as more significant than they really are. For example, if a count increased from 1 to 2 it would show up as +100%, whereas an increase from 5 to 6 would show up as 20%. Both counts are increasing by one, but as a percentage they seem to have different levels of significance. For this reason, the authors chose to leave these results as absolute counts and color-code the cells by the change magnitude.

17. Most of the analysis focused on the number of significant events... however, I think it would be helpful to describe how a QoI varies with a given parameter. Does it go up or down with respect to a parameter?

**Author Response:** The authors agree, and this is discussed and shown using Tables 6 and 7.

18. I found the discussion to be weak. The authors merely highlight the different results that the reader could easily see for themselves. it would really strengthen the work if the authors could give some insights and explainations on why the various parameters were significant. Why different changes in the loads were seen in the down stream turbines, etc.

    **Author Response:** The authors agree that this is a rich dataset and there is much more that can be learned from investigating it. However, given the already long length of the paper, the authors feel that the level and analysis is appropriate for the scope of work, and look forward to further investigating a smaller subset of the results.

19. I think more can be said in the conclusions. First, the fact that the sensitivity only changed by 3% when a turbine was within a wake is an important result in itself. Furtherore, the different in sensitivities between the loads is also interesting to mention. It's not clear to me how the authors classify primary, secondary and tertiary. Given the nature of the study, (i.e. high degree of aleatory uncertainty), it would be difficult to accurately identify low-order effects.

    **Author Response:** Additional details from the work have been added to the conclusions, particularly regarding the different sensitivities across QoI. Regarding the primary/secondary/tertiary classification, clarifying language about this distinction has been added to the text.

20. I think a very important result that is missing is an uncertainty quantification. Upon identifying the most important parameters, what would be my uncertainty in the loads if I was incorrect in my parameter value by a given value?

    **Author Response:** The authors agree that this could be an interesting and important aspect. However, a UQ analysis would require an extensive analysis and is beyond the scope of this paper.

**Reviewer #2**

1. It is not clear why TI is considered a QoI; it is neither a fatigue nor an ultimate load, although it is obviously related to them.

    **Author Response:** TI was added as a QoI because, as noted by the reviewer, structural loading is often driven by fatigue. The authors felt that including TI as a QoI provides this linkage to those who think of structural loading variations in terms of TI variations.

2. It should also be explained why a reference wind velocity is not included explicitly as an inflow parameter.

    **Author Response:** Instead of treating wind speed as a QoI, the sensitivity analysis is analyzed separately at three wind speed bins, characterizing below rated, at rated, and above rated operation. This approach was taken because wind speed variation is already considered

within design standards (and so is not uncertain or variable as the other parameters are). Moreover, three wind speed bins are considered (characterizing below rated, at rated, and above rated operation) because of the very different operational behavior of the wind turbine/controller across these conditions.

3. In line 146, I suppose that the range of variation of Δ is that specified in tables 3 and 4. It may convenient to justify better how these ranges have been obtained; in lines 173 to 181 some references are just quoted, however a brief summary evaluating the reliability of the cited work and of the range values proposed may be also of interest.

**Author Response:** These details are given in the previous paper related to this work. Due to journal standards regarding self-plagiarism this information was not repeated in this article, but another reference has been added to the previous work.

4. It is not clear whether figure 5 and table 5 refer only to blade root pitching moments as in figure 4, or to all Qoi…

**Author Response:** This figure and table refer to all QoI. This point has been clarified in the text and figure captions.

5. I cannot find specific comments in the text for Tables 5, 6 and 7, although their interpretation may be obvious. However, in tables 6 and 7 for WT2 and WT3, it looks as though percent differences (as indicated in eq. 5) are presented instead of the number of significant events. I just wonder if it would not be better to give also the number of significant events in those tables, as indicate the table captions.

**Author Response:** These tables were originally listed as Figures. The text was not updated when they were changed to being listed as Tables. Tables 6 and 7 show the number of significant events. For WT1 it is the absolute value and for WT2 and WT3 is it the count relative to the count of WT1. This has been clarified in the table captions.

Other comments:

6. Explain better the terminology: primary, secondary, tertiary, and if it means anything besides an order of relevance.

**Author Response:** Clarifying language about this distinction has been added to the text.

7. In your caveats about limitations, I think that the scarce number of turbines should also be mentioned.

**Author Response:** The authors assume this is referring to including 3 aligned wind turbines as opposed to a larger wind farm. This has been added to the list of caveats.

8. I suppose that figure 6 is an excedance histogram, similar to figure 4

**Author Response:** Figures 6 and 7 are actually histogram plots that show the number of times a certain Elementary Effects value was produced for a given QoI by a simulation, as is described in the text on page 13 line 261..

---

## Referee Report (RR1)

The revised manuscript has addressed many of the points in the first review. The main weakness in the first review was the quality of the presentation. Many of the points of confusion, missing information etc. has been adequately addressed. It was more pleasant reading this version and it was easier for me to interpret many of the plots. So I appreciate the authors second effort in improving the manuscript.